# Health Care and Access to Quality Social-Health Services of the Roma and Sinti: A Scoping Review

**DOI:** 10.3390/ijerph22071063

**Published:** 2025-07-02

**Authors:** Danilo Buonsenso, Davide Pata, Francesca Raffaelli, Giorgio Malorni, Francesca Colaiaco, Walter Malorni

**Affiliations:** 1Department of Woman and Child Health and Public Health, Fondazione Policlinico Universitario A. Gemelli IRCCS, 00168 Rome, Italy; 2Center for Global Health Research and Studies, Istituto di Igiene, Università Cattolica del Sacro Cuore, 00168 Rome, Italy; walter.malorni@unicatt.it; 3Dipartimento di Scienze di Laboratorio e Infettivologiche, Fondazione Policlinico Universitario A. Gemelli IRCCS, 00168 Rome, Italy; 4Bewweb srl, 00144 Rome, Italy; 5Italian Red Cross Association, 00151 Rome, Italy

**Keywords:** Roma, Sinti, public health, disease prevention, chronic diseases, access to cures, inequalities, minorities

## Abstract

Background: The aim of this scoping review is to analyze the health status of Roma and Sinti in Europe, highlighting the issues faced by children and women. In addition, we want to examine the access of these groups to health care services and to identify possible interventions to increase their use. Methods: Our research was conducted on Pubmed, Google Scholar, and the Trip Database. We selected articles written in English, Spanish, and Italian published since 2015. Results: Studies have shown that the health status of Roma and Sinti populations is generally worse than that of the rest of the population. Limited access to care is due to several specific factors, such as beliefs, traditions, and the lack of awareness of widespread direct and indirect discrimination against these groups by healthcare professionals. The studies reviewed have shown how mistrust can be broken down through multi-centered interventions linked to information, education, and communication through mediators able to interact with these populations, as well as through appropriate training of the health workers in charge. Conclusions: The health of the Roma and Sinti populations is commonly worse than that of the rest of the population. This is particularly true for the large proportion of people confined to suburban camps. However, the available evidence signals the low quality of life they experience and the need for interventions involving the communities and the establishment of ad hoc orientation or initial care contact points in the segregated areas. This could lead to an improvement in the integration of this population into the National Health Systems’ activities.

## 1. Introduction

The Roma are believed to be the largest ethnic minority group in the European region [1]. Generally, many populations, mostly concentrated in Central and Eastern Europe, such as the Sinti, Gypsies, Walkers, and Nomads, are included under the term Roma. The use of a single term, Roma, to refer to different groups does not reflect a reality that is completely different, in which ethnic, historical, and cultural specificities belonging to the various groups can be well distinguished [2,3]. In its official documents, the EU itself uses the generic term Roma to refer at the same time to several different groups [4]. However, there are external elements common to all the ethnic groups involved, such as a socio-economic situation of exclusion, marginalization, vulnerability, and poverty. In recent years, many European countries have taken action in defense of the human rights of minorities, including the Roma and Sinti, in order to address inequalities encountered at various levels, including those related to health [3].

The aspects most studied in the literature were the following: health status as a specific problem of these groups, if present, and the resulting treatment interventions. These interventions encompass the conscious choice of voluntary access to quality care systems, which include prevention activities, access to care systems, possible integrative interventions, and actions addressing the cultural and social context.

Surveys show that the health status of the Roma population is generally worse than that of the rest of the population, with a life expectancy for men of 7.5 years and for women of 6.6 years less than that of the general population. These figures vary according to the territorial areas considered, and can be as much as 15 years less [5,6]. This condition is present in the entire population, including women and children. Particularly, regarding the latter, important aspects related to birth planning, contraception, pregnancy, and both birth and post-birth conditions have been identified [7,8,9].

In addition, aspects related to primary prevention, such as screening and vaccinations, are deficient due not only to the convictions and traditions present in the cultures of these groups, but also due to the awareness of the presence of prejudice, stereotypes, and discrimination on the part of the health workers themselves towards them [10,11,12,13]. Although this aspect may seem secondary and relegated to segregated situations, the COVID-19 pandemic has highlighted how widespread diseases cannot be addressed without involving minority groups to prevent problems for the whole community. After all, the same problem had arisen previously in the context of measles outbreaks [14,15]. As a result of these considerations, the attitudes and impressions of the Roma and Sinti towards illness were analyzed alongside their approach to treatment and health care in order to understand the reasons why their access is limited, as well as the possibility of improving this situation through interventions with both top-down and bottom-up approaches [6,16]. Studies were then reviewed that highlighted the greater exposure of these groups to certain diseases, from infectious to parasitic, not only because of their settlement environment, which is characterized by generalized housing difficulties across Europe, but also because of possible specific genetic predispositions [17,18].

### Scoping Review Questions

What is the general health status of the Roma and Sinti populations?What are the main pathologies?What is the health status of women and children within the Roma and Sinti populations?Do these populations have access to health services?What interventions can improve the health status of the Roma and Sinti?

## 2. Materials and Methods

We performed a scoping review of the existing literature in accordance with the suggestions of the PRISMA statement [19].

### 2.1. Data Research

Our literature search strategy was conducted using PubMed, Google Scholar and the Trip Database (an online review of major scientific journals and official documentation from international, national, and local governmental or institutional organizations). The research was performed from January 2015 to June 2023, using the MeSH words “Roma” or “Sinti” in combination with “Health”, “Professional Care”, “Children”, “Women”, “health interventions and outcomes”, and “social determinant of health”.

### 2.2. Inclusion Criteria

Only studies investigating and including the health status of Roma and Sinti populations, as well as their access to quality health care services, were included, even if they were non-exclusively focused on these aspects. Studies and projects carried out and concluded with the aim of analyzing types of care, access to quality care services, and the evaluation of interventions to encourage access to health care services in Europe were also included in our analysis. All types of conclusive studies (qualitative, quantitative, interventional) were selected, as well as reviews of studies and editorials. Only articles written in English, Italian, and Spanish were considered.

### 2.3. Exclusion Criteria

Studies carried out before 2015 or that were inconclusive, not written in English, Spanish, or Italian, focused on different groups (such as nomads, gypsies, etc.), or were anthropological and historical articles were excluded.

### 2.4. Data Extraction

Two editors—specifically, first individually and then together—followed a protocol to analyze the selected documents in terms of general information, methodology, themes, and findings or conclusions.

### 2.5. Compliance with Ethics Guidelines

This scoping review is based on previously conducted studies, articles and institutional documents, and editorials.

## 3. Results

We initially imported 156 studies (Figure 1). A total of 62 of these were eliminated after an evaluation of titles and abstracts. After reading of the full text, a further 32 manuscripts were excluded: 20 were published before 2015, 2 were written in languages other than English, Italian, or Spanish, 4 were not conclusive, 2 concerned other populations, and 4 were duplicated.

The total number of works considered is 62, visible as a sum of the papers shown in Table 1 and Table 2 (see below).

A total of 36 studies focused specifically on Roma health (Table 1), 2 of which specifically included the Sinti. Out of the 36 studies, 17 were on women’s and pediatric issues and 7 were on possible genetic correlations. Other studies focused on other types of diseases, such as measles, diabetes, parasitic diseases, hepatitis, and mental health.

A total of 10 records constituted qualitative and quantitative surveys and research, while 5 were reviews on the condition of the Roma and Sinti, 3 analyzed aspects of anti-Roma racism, 3 examined traditions and cultural aspects related to health, and 5 featured recommendations of different agencies (Table 2).

The studies and research included covered various countries (Figure 2); in order of frequency: Slovakia (11), general Europe (11), United Kingdom (5), Spain (4), Bulgaria (3), Italy (3), Serbia (3), Hungary (3), Croatia (2), Wales (2), Ireland (2), Czech Republic (2), Greece–Macedonia (2), Finland (1), Portugal (1), and Slovenia (1).

### 3.1. Roma and Sinti Health

Several studies reported that the health of these groups is generally worse than the rest of the population, with a lower life expectancy. In particular, Roma men have a life expectancy of 7.5 years and Roma women of 6.6 years less than the general population [5,6].

A systemic review of water and hygiene among Roma communities in Europe found that most Roma populations experience worse health and well-being, including higher rates of communicable and non-communicable diseases, poorer child and maternal health, and higher mortality rates [20].

#### 3.1.1. Infectious Diseases

Segregated Roma communities are more affected than others by communicable diseases such as measles, hepatitis, and tuberculosis, have lower vaccination coverage, and are less likely to interact with health services [20,21,22].

Infectious diseases such as diarrhea, typhoid, hepatitis, scabies, and tuberculosis are more prevalent among the Roma than the rest of the population [9,18,20].

In addition, numerous unregistered animals without veterinary care live in many of their settlements and share living areas with people, so zoonoses such as cryptosporidiosis, microsporidiosis, toxoplasmosis, giardiasis, and infections with soil-borne pathogens are common [17,18,20].

Also, behavior and habits within settlements lead to increased rates of hepatitis B and E infection [17,20]. A Slovakian study showed that hepatitis B is high, and that mother-to-child transmission is probably one of the main modes of transmission and infection in early childhood based on low vaccination rates. Furthermore, fecal-oral and/or food-borne hepatitis (HAV and HEV) is more common in the Roma than in the non-Roma population. In contrast, HCV infection does not appear to be more frequent [17].

Both an Italian study and a French study also reported that the low vaccination coverage and the rapid and greater spread of settlements lead to a higher incidence of measles infection [11,23]. The COVID-19 pandemic has highlighted a similar problem, as the low vaccination coverage of these groups has led to outbreaks [14,24,25].

#### 3.1.2. Metabolic Diseases

A Finnish study found that 1 in 4 men and 1 in 3 women among the Roma had diabetes, while 1 in 3 men and 1 in 4 women over the age of 55 had coronary heart disease [26]. In this context, a Hungarian study found that Roma children regularly (at least five times a week) consumed sweets (11-year-old boys 66.5%, 11-year-old girls 71.4%; 13-year-old boys 69.5%, 13-year-old girls 75.7%) and soft drinks (11-year-old boys 77.9%, 11-year-old girls 71.2%; 13-year-old boys 80.3%, 13-year-old girls 80.7%) [27]. In relation to complex diseases, a Slovakian analysis shows a higher prevalence of type 2 diabetes (T2DM) and pre-diabetes (PreDM), metabolic syndrome, cardiovascular disease, obesity, overweight, and hypertension [17].

#### 3.1.3. Women and Children’s Health

A Bulgarian study found that most Roma women start having sex at the age of 14 (20.58%), followed by those at 15 (14.70%) and 16 (13.23%). The average age was 16. Only 21.73% of the sample used contraceptives regularly, while 39.13% of the women did not use them and 39.13% rarely used them [7].

A retrospective study of Slovakian medical records also found that the low average age at the start of sexual life and the consequent low age of women at first childbirth lead to abortion and premature birth in many cases [17]. In addition, a Slovenian study on the reproductive health of Roma women indicated that several similar obstacles can be encountered practically all over Central and Eastern Europe, such as lack of financial means, difficulties in obtaining health insurance, and dependence on formal and informal forms of payment when required, as well as fear regarding their expectation of a gynecological examination. However, when well-informed, women voluntarily accessed the gynecological clinic [28].

A Serbian study found that obligatory virginity for girls before their first endogamous marriages and high fertility rates lead to teenage marriages and births. According to their estimates, almost 40 per cent of Roma women gave birth to their first child before the age of 18, 10% became mothers before the age of 16 and, among women aged 15–49, 16% gave birth to a first child before the age of 15 [29].

An Irish review found a higher rate of hereditary diseases among Roma children due to the practice of intermarriage in some clans [5]. Similarly, a Slovak study found that Roma children represent 24.2% of all cases of sudden infant death syndrome in Slovakia, which is much higher than the estimated proportion of Roma children in the general population.

Vision correction in a Bulgarian ad hoc study was found to be a traditionally uncommon practice even among adults. The possibility of having access to examinations and the provision of free spectacles certainly encouraged the parents involved in the study to consider and pay attention to this aspect as well [30].

Regarding oral hygiene, a Slovakian survey divided children into two groups, those with primary dentition (6 years) and those with permanent dentition (12 years). Dental examinations were carried out in each group. No radiographs were taken. The study revealed that 46.5% of 6-year-old Roma children brush their teeth without toothpaste, with 172 in the group of 12-year-olds having the most common response that they brush their teeth once a day. In the study, the results showed that 6-year-old Roma children had a statistically significantly higher incidence of dental caries (*p* < 0.05) [31].

The prevalence of acute malnutrition or emaciation was 4.8% versus 1% in the counterpart population, while the prevalence of stunting was 21.4% versus 2%. The prevalence of underweight was 14.2% versus 1% [9].

A Serbian study highlighted that Roma children born to educated mothers and living in households with access to improved sanitation were better nourished than their peers with uneducated mothers living in poor conditions [29].

The eating habits of the Roma children were very unfavorable in terms of the consumption of sweets and soft drinks. In a Hungarian study, although about two-thirds of the children consumed fruit every day, more Roma girls at the age of 11 reported less fruit consumption. A large difference was observed between the two population groups in the consumption of sweets and soft drinks. Among Roma children, the prevalence of regular consumption (at least five times a week) of sweets (boys aged 11 years 66.5%, girls aged 11 years 71.4%; boys aged 13 years 69.5%, girls aged 13 years 75.7%) and soft drinks (boys aged 11 years 77.9%, girls aged 11 years 71.2%; boys aged 13 years 80.3%, girls aged 13 years 80.7%) were one and a half and two times that of the general population [32]. On the other hand, a Spanish study pointed out that the Roma population is in an unequal situation regarding levels of adherence to the healthy eating guidelines proposed by the Spanish national strategy. The distance to these healthy eating habits is even greater among the younger Roma population [33].

#### 3.1.4. Genetic Predisposition

The COL4A4 pathogenic variant causes autosomal recessive Alport syndrome, which is responsible for most genetic kidney failures in the Roma population [17,34]. A 2021 study found a pathogenic variant of the CHEK2 founder in Portuguese Roma as a likely basis of thyroid cancer and other tumor manifestations in the Roma population [35]. Furthermore, in view of the early onset and high prevalence of smoking in these groups, one study considered the possibility that smoking addiction, which is prevalent in the Roma population, may have a genetic predisposition. This hypothesis, which assumed that individuals were genetically susceptible to smoking, was rejected, and instead elements related to tobacco control policies rather than genetics were found [36].

A Slovakian paper on uric acid showed that serum uric acid is gender and ethnic specific. A comprehensive analysis of the data revealed significantly elevated levels of C-reactive protein (CRP) and serum ferritin, particularly in Roma men, which may reflect low-grade systemic inflammation and thus serve as an indicator of increased cardiovascular risk. The significantly lower serum uric acid levels, together with positive linear associations with albumin, may reflect the reduced antioxidant status of the Roma population [37]. Another Spanish study showed that the Roma are not so genetically isolated. Gene flow with other European groups accounts for 65% of their genetic ancestry, so any clinically harmful variants are traced back to both South Asian and European haplotypes [38].

Similarly, a Slovakian study of dialyzed persons found that at the cross-sectional level, when adjusted for age and gender, no significant differences were identified in the health status of dialyzed Roma and non-Roma in Slovakia. However, the younger age of dialyzed Roma and their shorter stay on dialysis suggest a higher mortality among Slovakian Roma patients [39].

#### 3.1.5. Cultural and Mental Aspects

A Slovak participatory study conducted in 2018–2019 found that participants described cancer as a taboo disease that was not talked about outside the family and was even hidden from others in the community [13]. Referring to organ donation, most of the Roma population surveyed in a Spanish sociological study preferred not to talk about death or organ donation after death [40]. Diseases such as cancer are not only avoided, but often diagnosed late due to poor access to screening systems, as highlighted in a qualitative study in Slovakia [13].

Attitudes towards death and respect for the dead do not differ significantly from those of the majority of the population. Dying at home is less common than in the past, while dying in hospital is more frequent [13,40].

A study of immunization implementation strategies in the United Kingdom highlighted how information about immunization was received from within the community or through social media [12].

In a study in Eastern Macedonia and Thrace, Roma were assessed using the Derogatis Psychiatric Rating Scale. It was found that Roma patients were more likely to present with stress disorders, somatization, conversion symptoms, and agitation, usually with a histrionic and attention-seeking background when suffering from depression or anxiety disorders. They respond moderately to treatment and their progress is slow and unstable; they tend to relapse more often and their overall stress is resistant to treatment. The percentage of Roma women attending care centers was higher than that of Roma men and higher than that of women in the majority population. Their symptom burden and general distress were also measured to be higher than that of Roma men [41]. A Finnish study also found that the prevalence of mental illness, particularly depression, appears to be high among Finnish Roma [26].

### 3.2. Access to Care and Treatment Systems

A Finnish study highlighted how the awareness of being indirectly discriminated against influences their attitudes and behavior [26]. It is quite common for minority groups to experience direct and indirect discrimination when interacting with care systems [12,26,42]. Other problems of access remain common:Mobility (difficulty in reaching contact points or cost of travel)Cost of medicines and treatments, if not free of chargeLanguage problems, illiteracy, and misinformationMistrust of health workersPrevious negative experiences of health careDisinformation

The presence of a pathology and the seeking of any necessary treatment involves not only a socio-health aspect, but also a cultural aspect. As highlighted in the previous discussion, the health of Roma and Sinti is poorer and less well cared for in all European countries. Although social, economic, and political conditions have a great influence on this aspect, it is also true that some studies have also analyzed the educational sphere of these groups in order to understand and interpret the intentions behind their behavior. An important aspect is the understanding and observance of norms, including those related to health. A Slovakian study has shown a number of aspects, one of which is that isolated groups have a tendency to create very strong group laws, which become even stricter if the group remains isolated. In such cases, group laws prevail to the detriment of social laws [43]. Non-compliance may also depend on an assimilation mechanism of the same racist anti-Roma ideologies, often accompanied by fatalism about possible outcomes [26]. Again, in a Slovakian study, the family group compensates for the lack of access to care in case of difficulties in obtaining it [27].

### 3.3. Possible Actions

A welcoming and non-judgmental approach at first contact, with the help of facilitators and peers, has been found to be very effective [5,44]. Establishing mobile and exclusive access points, at least at an early stage, was also found to be an effective way of promoting counseling services and providing clear and useful information [5,26,42]. Another successful intervention was found to be the provision of flexible booking systems for appointments, with targeted invitations to prevention and subsequent reminders. This aspect has also been linked to the difficulty of counting these groups [5,12,26].

Workshops facilitated by health workers for some recruited members of the Roma and Sinti communities (or “facilitators”) allowed the creation of a link [16]. Having a point of reference for specific needs within the community who can also be a link has proven to be a good incentive [16]. Some participatory studies have shown that socio-economic issues related to housing and livelihood are often the primary and main concern, neglecting fundamental issues such as health and all the individual attitudes and behaviors that influence it [13]. One study considered the possibility of encouraging disadvantaged groups to participate in qualitative research through economic benefits, preferring instead incentives such as vouchers [45].

## 4. Discussion

The studies analyzed highlighted aspects related to two main themes: Roma and Sinti health and care and access to quality care systems. The ultimate goal was to organize programs, studies, and interventions in order to promote and improve trust and communication between these communities and the care system, including prevention and vaccination activities for the ethnic groups investigated.

### 4.1. Roma and Sinti Health

The Roma population is one of Europe’s most marginalized groups, with estimates suggesting a population of around 12 million in Europe [46].

A substantial portion of this group lives in socioeconomically disadvantaged conditions, both in rural and urban settings. These marginalizing circumstances, including segregation and poverty, expose the Roma to greater vulnerability, particularly in terms of access to healthcare. Studies, such as those conducted in Slovakia [43], emphasize that these conditions contribute to lower health outcomes, heightened mortality rates, and reduced life expectancy when compared to the general population [5,6].

Environmental factors, particularly poor housing and living conditions, remain central to Roma health issues. Substandard living conditions, poor nutrition, and higher exposure to harmful substances (alcohol, tobacco) directly impact overall well-being and health [22,36]. Studies suggest these environmental factors are key contributors to the higher prevalence of several diseases, including metabolic syndromes, which may be exacerbated by ethnic factors, although further research is needed [17,23,47].

Additionally, cultural barriers such as language differences and discriminatory attitudes from healthcare providers further complicate access to care. The Roma and Sinti often experience both direct and indirect discrimination, which, compounded by limited mobility and economic barriers, isolates them from the formal healthcare system [37].

#### 4.1.1. Infectious Diseases

Research indicates that Roma living in segregated settlements—often located near industrial zones, landfills, or agricultural cooperatives—are at heightened risk of infectious diseases due to inadequate water and sanitation facilities. The presence of waste, rodents, and insects fosters an environment ripe for the transmission of waterborne diseases such as typhoid, hepatitis, and diarrheal diseases [9,18,20]. Roma children, in particular, are disproportionately affected by parasitic infections and zoonoses, underlining the severity of health risks in these communities [17,18,21].

Viral hepatitis, in addition, is a significant concern, with Roma populations exhibiting elevated vulnerability due to poor living conditions and limited access to medical care. Measles outbreaks in these communities, largely a consequence of low vaccination coverage, further highlight the gaps in public health interventions. The COVID-19 pandemic underscored the challenges of achieving high vaccination rates among these groups, exacerbating the already fragile healthcare infrastructure in marginalized communities [14,24,25].

#### 4.1.2. Metabolic Diseases

Metabolic conditions, such as diabetes, are notably prevalent within Roma populations, though evidence of a genetic predisposition remains inconclusive. Systematic reviews have yet to provide definitive answers due to the lack of targeted studies focusing on this aspect [47]. However, the significant role of socio-economic and behavioral factors, including poor diet and limited access to preventive healthcare, appears to be a primary driver of the high incidence of these conditions in Roma communities.

#### 4.1.3. Women and Children’s Health

Research on Roma women’s health is limited, with a European Commission report [46] acknowledging the need for more focused studies. The existing literature suggests that Roma women face particular challenges related to reproductive health [7,8,9,48]. Early marriages and pregnancies, coupled with limited access to prenatal and maternal care, result in poorer health outcomes for both mothers and children [8,32]. Pregnant Roma women living in poverty are at a higher risk of complications such as preterm birth and pre-eclampsia, exacerbating their vulnerability [49].

In relation to breastfeeding, a woman’s decision to exclusively breastfeed is associated with the duration of lactational amenorrhea. In the general population, women’s decisions to exclusively breastfeed are strongly associated with other factors, such as the type of settlement, the age of their partner, and the fact that the baby has had something other than breast milk in the maternity ward [48].

Children are also susceptible to environment-related diseases from birth. For example, Giampaolo et al. described a high percentage of underweight Roma children living in camps [9].

The desire for children emerged as a predictor of breastfeeding practices and vaccination status: Roma mothers directed their investments towards desired children. The quality of mother–child interactions varied according to the wealth of the mother’s household [50].

Marginalized Roma children also suffer more from infectious diseases, injuries, poisoning, burns, respiratory diseases, and chronic illnesses than other children [42].

Another aspect concerns the correction of refractive error. Vision problems clearly affect quality of life, and in children also have an impact on school performance [30].

Similarly, a Slovakian survey highlighted that oral hygiene is insufficient in the Roma children’s population, particularly in the 6–12 year-old age group, and that information programs in this regard would be necessary [31]. A specific problem for school-age children is an excessive intake of sugary drinks and the consumption of sweets, which leads to a caries process [31].

Sárváry et al. highlighted that children living in Roma settlements reported worse socio-economic conditions, higher consumption of sweets and soft drinks, earlier initiation into smoking and alcohol, and worse self-assessment of health. However, with some exceptions, they did not differ in fruit or vegetable consumption and BMI compared to the general children’s population. The study concluded that a multi-sectoral approach, specific health education, and social and health promotion programs are needed to promote the health of children living in Roma settlements [32].

#### 4.1.4. Genetic Predisposition

While some studies have highlighted the prevalence of certain diseases in the Roma population, such as metabolic disorders, kidney issues, and cardiovascular diseases, the evidence linking these health conditions to genetic predispositions remains inconclusive. Some studies suggest that the higher incidence of these diseases may be influenced by the Roma’s history of social isolation and intermarriage, which may contribute to the recurrence of specific genetic conditions [17]. However, ethical concerns regarding the collection and use of genetic data in marginalized populations must also be considered [51].

#### 4.1.5. Cultural Aspects

Cultural beliefs significantly shape health outcomes within the Roma community. Illness is often viewed through a cultural lens, where serious conditions may be seen as signs of weakness or even supernatural causes [13,40]. This cultural framework, coupled with limited access to formal education, contributes to misunderstandings about health and wellness, leading to lower levels of health literacy [12]. Mental health issues, particularly among Roma women, also seem to be underreported and inadequately addressed, further compounding health disparities [26,41].

### 4.2. Access to Care and Treatment Systems

A range of studies and participatory surveys have examined the dynamics of cooperation between Roma and Sinti communities and institutional healthcare systems, shedding light on the key barriers to effective access to care. One of the most prominent findings is the widespread mistrust these populations have toward healthcare services—not only in terms of the efficacy of treatments, but also with regard to the quality of care and the relationship with healthcare professionals and institutions [52].

The most significant obstacle identified is the strained relationship between healthcare providers and Roma patients, often exacerbated by language barriers and cultural differences. These issues lead to communication breakdowns, which are compounded by perceptions of discrimination—both direct and indirect—at healthcare access points. The lack of culturally competent staff remains a persistent problem across European contexts. Moreover, in certain countries, the compulsory nature of health insurance or other forms of cost-sharing further discourages Roma individuals from seeking care [5]. These barriers contribute to the ongoing marginalization of the Roma and Sinti, which, in turn, limits the ability of healthcare systems to effectively address their needs. Additionally, issues such as census challenges and limited social and cultural integration hinder accurate assessments of their health conditions, further complicating efforts to provide targeted care.

### 4.3. Possible Actions

The reviewed studies provide valuable insights into interventions that could improve healthcare access for the Roma and Sinti populations, identifying several key areas for action. One of the most critical aspects is addressing communication barriers between health workers and minority patients. Often, healthcare providers lack the necessary training to navigate cultural differences or effectively communicate with Roma patients, which undermines trust in the system. Thus, training healthcare professionals in Roma and Sinti cultures, including gender dynamics, is a critical step toward improving the quality of care [26].

Additionally, participatory approaches have proven effective in overcoming barriers to care. Programs that engage both healthcare workers and community members have shown promise in bridging the gap between these groups. Workshops that allow patients to express their concerns and needs directly foster a sense of partnership, while the use of “facilitators” or community mediators helps address specific cultural and linguistic challenges [16,53]. These initiatives not only enhance communication but also strengthen the sense of ownership and empowerment within the community.

It is important to highlight that the European Union has committed to reducing health inequalities, particularly those affecting marginalized groups like the Roma and Sinti. In alignment with this goal, national guidelines have been issued to encourage more inclusive and empathetic healthcare delivery. These policies aim to foster virtuous behaviors among healthcare workers and the broader society, promoting more equitable access to health services for vulnerable communities [44,54,55].

## 5. Conclusions

This scoping review provides a comprehensive overview of the health disparities experienced by Roma and Sinti populations in Europe, emphasizing the multifaceted barriers to healthcare access and the interventions that could address these challenges. Overall, Roma and Sinti face worse health outcomes than the general population, largely due to their confinement in marginalized settlements, which often lack basic infrastructure. Infectious and parasitic diseases are prevalent, with environmental factors—such as poor housing and sanitation—playing a central role in their spread. Additionally, cultural practices and social norms, such as early marriages, smoking, and low vaccination rates, contribute to unhealthy behaviors and exacerbate health risks.

Chronic conditions, such as diabetes, cardiovascular diseases, and hypertension, are widespread among these populations and are often linked to unhealthy lifestyles, including poor diet, alcohol consumption, and smoking. For women and children, the lack of prenatal and postnatal care, combined with higher rates of malnutrition and premature birth, results in negative health outcomes that disproportionately affect this demographic. The studies reviewed suggest potential genetic predispositions to certain diseases (e.g., metabolic disorders, kidney diseases, thyroid cancer), but the evidence remains inconclusive due to the limited number of studies and the heterogeneity of results.

A consistent theme across the reviewed literature is the deep mistrust Roma and Sinti populations have toward healthcare systems. This mistrust is fueled by language barriers, perceived discrimination, misinformation due to low educational levels, and social isolation—all of which hinder effective participation in healthcare programs. Furthermore, factors such as the absence of compulsory health insurance, mobility challenges, and the complexity of conducting accurate censuses further exacerbate access issues.

Research on interventions designed to improve healthcare access consistently suggests that training healthcare professionals to be culturally competent and non-judgmental, alongside the use of community-based mediators, is crucial for building positive relationships between the Roma and healthcare providers. Action-research interventions that involve communities in identifying and addressing their healthcare needs have proven effective in fostering better engagement and improving health outcomes. The establishment of accessible contact points, such as health orientation centers near marginalized areas, also contributes to the creation of supportive networks that facilitate better healthcare access.

This review underscores the importance of addressing both the systemic barriers and the cultural dimensions that shape healthcare access for Roma and Sinti communities. While the studies reviewed are often limited by small sample sizes and the variation of issues across countries, they collectively highlight the need for more inclusive and context-specific interventions. The contradictions in findings across studies further underscore the need for more targeted and comprehensive research to understand the full scope of health disparities faced by these populations.

## Figures and Tables

**Figure 1 ijerph-22-01063-f001:**
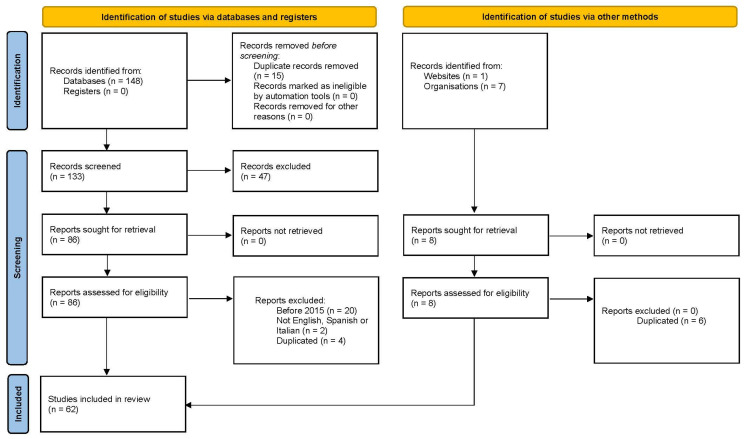
From: Page MJ, McKenzie JE, Bossuyt PM, Boutron I, Hoffmann TC, Mulrow CD, et al. The PRISMA 2020 statement: An updated guideline for reporting systematic reviews. BMJ 2021;372:n71. [19]. For more information, visit: http://www.prisma-statement.org/.

**Figure 2 ijerph-22-01063-f002:**
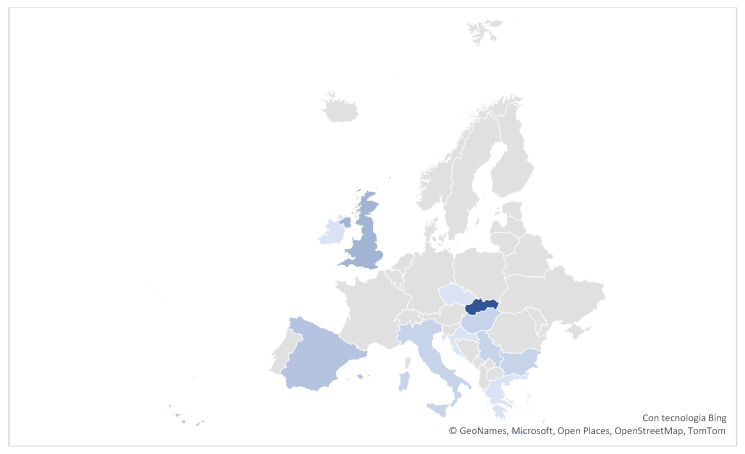
Map with the origin of the studies.

**Table 1 ijerph-22-01063-t001:** **Topics of studies included in the review**.

Studies on women and pediatric issues	17
Genetic investigation studies	7
Dialysis studies	1
Idiopathic arthritis studies	1
Diabetes studies	2
Hepatitis studies	3
Parasitic disease studies	2
Mental illness studies	1
Measles studies	2
Total health status studies	36

**Table 2 ijerph-22-01063-t002:** **Different types of studies included in the review**.

Surveys and Research	10
Systemic reviews	5
Racism	3
Traditions and culture	3
Institutional Recommendations/Publications	5
Total	26

## Data Availability

No new data were created or analyzed in this study.

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
