# Peer review of "Health Care and Access to Quality Social-Health Services of the Roma and Sinti: A Scoping Review"

_ijerph, 2025, doi:10.3390/ijerph22071063_

Round 1
Reviewer 1 Report
Comments and Suggestions for Authors
The scoping review is interesting, but I suggest to improve the work developed because is very "didactic" (form my personal point of view) and it can be more and more well-argued and the conclusions can suggest possible perspectives.
Do you consider the topic original or relevant to the field? Does it
address a specific gap in the field?
The paper is very specific and useful for specific colleagues of Scientific Community expert on the filed of interest.
What does it add to the subject area compared with other published material?
The scoping review -in that sense-take in consideration very punctual works.
What specific improvements should the authors consider regarding the methodology?
The methodology works well. My personal opinion the Prisma graph lack of the selection of Pubmed / Google Scholars / Trip database databases. Nowhere is clever how many works did you find from the 3 databases, and how many are "in common".
In general, I think that the discussion is to much didactic. Every paragraph argues the outcomes of each paper... but probably the texts can be better written and argued. "A finish/bulgarian/French/... study" are very boring. You can referent o the authors and their affiliation... or something else.
In that way the paper can be more interesting, but in present form is very didactic.
Are the conclusions consistent with the evidence and arguments presented and do they address the main question posed?
Yes, they work but probably you can better argue.
Are the references appropriate?
yes, they are. Please have a check to the reference style of n. 46 and -my personal suggestion- to improve the n.20 adding with round brackets the English translation of the Italian document argued.
In conclusion, just for your information, for the question "Is the work a significant contribution to the field?" I answer 3/5 because the topic is too much specific and probably the experience on Rome context can be related to a limited number of colleagues of the Scientific Community.
Author Response
Dear Reviewer,
Thank you very much for your efforts aiming to improve our paper.
Please find below a point-by-point response to your comments. Changes have been highlighted in yellow in the main manuscript.
The scoping review is interesting, but I suggest to improve the work developed because is very "didactic" (form my personal point of view) and it can be more and more well-argued and the conclusions can suggest possible perspectives.
Do you consider the topic original or relevant to the field? Does it
address a specific gap in the field?
The paper is very specific and useful for specific colleagues of Scientific Community expert on the filed of interest.
What does it add to the subject area compared with other published material?
The scoping review -in that sense-take in consideration very punctual works.
Thank you, we share your thoughts and we modified our manuscript as suggested
What specific improvements should the authors consider regarding the methodology?
The methodology works well. My personal opinion the Prisma graph lack of the selection of Pubmed / Google Scholars / Trip database databases. Nowhere is clever how many works did you find from the 3 databases, and how many are "in common".
In general, I think that the discussion is to much didactic. Every paragraph argues the outcomes of each paper... but probably the texts can be better written and argued. "A finish/bulgarian/French/... study" are very boring. You can referent o the authors and their affiliation... or something else.
In that way the paper can be more interesting, but in present form is very didactic.
Are the conclusions consistent with the evidence and arguments presented and do they address the main question posed?
Yes, they work but probably you can better argue.
We rewrote the entire "Discussion" and "Conclusion" sections as suggested. Thank you, we believe the manuscript is improved now
Are the references appropriate?
yes, they are. Please have a check to the reference style of n. 46 and -my personal suggestion- to improve the n.20 adding with round brackets the English translation of the Italian document argued.
Corrected as suggested
In conclusion, just for your information, for the question "Is the work a significant contribution to the field?" I answer 3/5 because the topic is too much specific and probably the experience on Rome context can be related to a limited number of colleagues of the Scientific Community.
Thanks for your comment
Thank you for helping us improve our manuscript.
Davide Pata
On behalf of all co-authors
Reviewer 2 Report
Comments and Suggestions for Authors
Brief Summary
This scoping review aims to analyze the health status of Roma and Sinti populations in Europe, examine barriers to healthcare access, and identify effective interventions to improve healthcare utilization. The study synthesizes evidence from 62 studies across 15 European countries, providing a comprehensive overview of health disparities, access challenges, and potential solutions for these marginalized communities. The manuscript addresses an important and understudied topic with significant policy implications for European health systems.
General Concept Comments
Strengths:
- Addresses health inequities in Europe's largest ethnic minority, filling a critical gap in the literature
- Includes evidence from 15 European countries, enhancing generalizability
- Scoping review approach is well-suited for mapping this broad and heterogeneous field
- Provides actionable insights for healthcare providers, policymakers, and public health practitioners
- Appropriately focuses on vulnerable subgroups including women and children
The research is generally well-conducted within the scoping review framework, and the findings have clear policy and practice implications.
Areas of Concern:
The title claims "Scoping Review AND Narrative Analysis" but the manuscript presents only a traditional scoping review. No narrative analysis methodology is described in the methods section, no theoretical framework for narrative analysis is provided, and no actual narrative analysis is conducted. This creates misleading expectations and reduces methodological credibility.
The stated aim contains confusing language: "to detect realities in which any interventions have led to an incentive to use these services" is grammatically unclear and methodologically imprecise. The manuscript lacks explicit research questions, which are essential for scoping reviews according to PRISMA-ScR guidelines.
Minor Issues
- Line 78: "Updated scoping review" - unclear what previous review this updates
Recommendations
- Either remove "Narrative Analysis" from the title or add actual narrative analysis methodology and results
- Provide clear, specific research questions in standard scoping review format
Author Response
Dear Reviewer,
Thank you very much for your efforts aiming to improve our paper.
Please find below a point-by-point response to your comments. Changes have been highlighted in yellow in the main manuscript.
Brief Summary
This scoping review aims to analyze the health status of Roma and Sinti populations in Europe, examine barriers to healthcare access, and identify effective interventions to improve healthcare utilization. The study synthesizes evidence from 62 studies across 15 European countries, providing a comprehensive overview of health disparities, access challenges, and potential solutions for these marginalized communities. The manuscript addresses an important and understudied topic with significant policy implications for European health systems.
Thanks for your comment
General Concept Comments
Strengths:
- Addresses health inequities in Europe's largest ethnic minority, filling a critical gap in the literature
- Includes evidence from 15 European countries, enhancing generalizability
- Scoping review approach is well-suited for mapping this broad and heterogeneous field
- Provides actionable insights for healthcare providers, policymakers, and public health practitioners
- Appropriately focuses on vulnerable subgroups including women and children
The research is generally well-conducted within the scoping review framework, and the findings have clear policy and practice implications.
Thanks for your comment
Areas of Concern:
The title claims "Scoping Review AND Narrative Analysis" but the manuscript presents only a traditional scoping review. No narrative analysis methodology is described in the methods section, no theoretical framework for narrative analysis is provided, and no actual narrative analysis is conducted. This creates misleading expectations and reduces methodological credibility.
Thank you, we changed the title as suggested
The stated aim contains confusing language: "to detect realities in which any interventions have led to an incentive to use these services" is grammatically unclear and methodologically imprecise.
We rephrased the sentence
The manuscript lacks explicit research questions, which are essential for scoping reviews according to PRISMA-ScR guidelines.
Thank you, we added the paragraph “scoping review question”
Minor Issues
- Line 78: "Updated scoping review" - unclear what previous review this updates
We eliminated “updated”
Recommendations
- Either remove "Narrative Analysis" from the title or add actual narrative analysis methodology and results
Performed
- Provide clear, specific research questions in standard scoping review format
Performed as suggested
Thank you for helping us improve our manuscript.
Davide Pata
On behalf of all co-authors
Round 2
Reviewer 2 Report
Comments and Suggestions for Authors
Thank you for your careful and thoughtful revisions. I appreciate the improvements you have made to the manuscript. All of my previous concerns have been adequately addressed, and I find the current version acceptable for publication. I wish you success in your future research.